# Anti-Tumor Secondary Metabolites Originating from Fungi in the South China Sea’s Mangrove Ecosystem

**DOI:** 10.3390/bioengineering9120776

**Published:** 2022-12-06

**Authors:** Yuyou Luo, Xiongming Luo, Tong Zhang, Siyuan Li, Shuping Liu, Yuxin Ma, Zongming Wang, Xiaobao Jin, Jing Liu, Xin Wang

**Affiliations:** 1School of Life Sciences and Biopharmaceutics, Guangdong Pharmaceutical University, Guangzhou 510006, China; 2Guangdong Provincial Key Laboratory of Pharmaceutical Bioactive Substances, Guangdong Pharmaceutical University, Guangzhou 510006, China; 3Pituitary Tumor Center, Department of Neurosurgery, The First Affiliated Hospital, Sun Yat-sen University, Guangzhou 510080, China

**Keywords:** secondary metabolites, mangrove, fungi, anti-tumor, South China Sea

## Abstract

A mangrove is a unique ecosystem with abundant resources, in which fungi are an indispensable microbial part. Numerous mangrove fungi-derived secondary metabolites are considerable sources of novel bioactive substances, such as polyketides, terpenoids, alkaloids, peptides, etc., which arouse people’s interest in the search for potential natural anti-tumor drugs. This review includes a total of 44 research publications that described 110 secondary metabolites that were all shown to be anti-tumor from 39 mangrove fungal strains belonging to 18 genera that were acquired from the South China Sea between 2016 and 2022. To identify more potential medications for clinical tumor therapy, their sources, unique structures, and cytotoxicity qualities were compiled. This review could serve as a crucial resource for the research status of mangrove fungal-derived natural products deserving of further development.

## 1. Introduction

A mangrove is a unique ecosystem distributed along the coastline of the tropical and subtropical regions, whose specific saline environment is what gives rise to its diverse microbial population [1]. Additionally, the biodiversity of fungi is more abundant than that of other mangrove microorganisms, which has a fairly good prospect of exploration that causes overwhelming focus [2,3]. Modern research suggests that the secondary metabolites which mangrove-derived fungi yielded are considerable sources of novel bioactive compounds [4]. The OSMAC strategy (one strain, many compounds) and co-cultures of mangrove fungus, as well as other microbes, have become the traditional methods for inducing the formation of metabolites [4,5]. The recent explosive multiomics methods wrestle with multi-trophic interactions and computational approaches based on artificial intelligence to estimate function distribution and demonstrate a paradigm change in the use of high-throughput methods for the discovery of natural products from plant-associated microorganisms or traditional Chinese medicine as the potential biomarkers for anti-cancer therapies [6,7]. Further research for these metabolites can be expected to discover their various biological features, such as enzyme inhibition, antimicrobial activity, antitumor activity, anti-inflammatory activity, antiviral activity, cytotoxicity, antioxidant activity, etc. Around the world, mangrove fungus-derived chemicals are being discovered at a progressively higher rate. Among them, many promising ones are beneficial in treating diseases [8]. 

Cancer, one of the leading causes of human death and the biggest impediment to extending life expectancy, is a serious worldwide health issue [9]. According to cancer statistics worldwide, there would be about 19.2 million emerging cases and 6.0 million cancer deaths occurring in 2022. In the past decade, anti-cancer research has accelerated due to rapid advances in early detection, surgical techniques, chemotherapy, radiation therapy, and targeted therapy [10]. Cytotoxic drugs are the mainstay of tumor chemotherapy, but their safety and resistance are unavoidable clinical concerns [11,12,13]. More novel anti-tumor drugs are still needed.

The South China Sea, a major mangrove region in the world, has been exploited and utilized for its rich mangrove fungal resources for many years, from which many promising potential anti-tumor drugs have been investigated for cancer treatment [8]. Xyloketal B originated from South China Sea’s mangrove fungal strain *Xylaria* sp. (No. 2508), and has displayed inhibition against glioblastoma cell proliferation and migration [14]. SZ-685C, an anthraquinone from another fungal strain *Halorosellinia* sp. (No. 1403), could suppress the reproduction of six human cancer cell lines and mice breast carcinoma xenografts [15]. Penicisulfuranol A, a new structural compound originated from *Penicillium janthinellum* possessing an infrequent benzofuran piperazine ring, was found to particularly suppress the C-terminal of Hsp90 (heat shock protein 90) and limit the development of HCT116, the human colon cancer cell line, not only in vitro but also in vivo [16]. These natural products, with their obvious anti-tumor effects and abundant candidate resources, have become the priority for anti-tumor drug development.

In this review, the sources, structures, and anti-tumor properties of the secondary metabolites isolated from fungal strains which were acquiredfrom the South China Sea’s mangrove ecosystem were outlined. A total of 44 research papers describing 110 anti-tumor compounds reported between 2016 and 2022 were included; the vast majority of those are novel compounds discovered in this period.

## 2. Summarisation of Anti-Tumor Secondary Metabolites from Mangrove-Derived Fungi

The one hundred and ten anti-tumor secondary metabolites obtained in the past seven years from fungi produced from the South China Sea’s mangrove ecosystem can be branched out into four types according to their chemical constructions and properties: polyketides (including nine groups: azaphilones, coumarins and isocoumarins, chromones, lactones, benzoates, xanthones, quinones and benzophenones, phenol, phenyl aldehydes, phenolic acids, and depsidones) (56, 51%), terpenoids (including three groups: sesquiterpenes, diterpenes, and steroids) (14, 13%), alkaloids (including three groups: amines and amides, diketopiperazines, and cytochalasins) (39, 35%) and peptides (including one group: cyclic peptides) (1, 1%) (Figure 1a).

These secondary metabolites were identified from the 39 strains belonging to 18 genera (*Penicillium*, *Aspergillus*, *Cladosporium*, *Diaporthe*, *Phomopsis*, *Fusarium*, *Colletotrichum*, *Lasiodiplodia*, *Mucor irregularis*, *Phoma*, *Sarocladium*, *Xylaria*, *Cytospora*, *Didymella*, *Eutypella*, *Pseudofusicoccum*, *Chaetomium*, and *Pseudopithomyces*). Among them, the compounds derived from *Penicillium* (26, 23%), *Aspergillus* (25, 23%), and *Lasiodiplodia* (12, 11%) account for more than half of the total secondary metabolites, illustrating that the three genera are the major producers of the anti-tumor compounds (Figure 1b).

## 3. Sources, Structures, and Anti-Tumor Activities of Secondary Metabolites Originating from Fungal Strains in the Mangrove Ecosystem

One hundred and ten anti-tumor compounds obtained from fungi produced from the South China Sea’s mangrove ecosystem were reported from 2016 to 2022 and their classifications, sources, and cytotoxicities (described with IC_50_, the half maximal inhibitory concentration) were listed (Table 1).

### 3.1. Polyketides

Polyketides make up more than half of the recently discovered anti-tumor secondary metabolites of mangrove-associated fungus from 2016 to 2022. They can be branched out into nine categories on the ground of the polyketide backbone, including azaphilones, coumarins and isocoumarins, chromones, lactones, benzoates, xanthones, quinones and benzophenones, phenols, phenyl aldehydes, and phenolic acids, as well as depsidones (Compounds **1**–**56**, Figure 2 and Figure 3).

#### 3.1.1. Azaphilones

Azaphilones, the major class of fungal polyketides, are known to possess the structural features of a highly oxidized pyranoquinone core [61]. In the past seven years, three azaphilones (**1**–**3**) with remarkable anti-tumor bioactivity were reported from the fungal genera *Diaporthe* sp. SCSIO 41011. They originated from the mangrove and were found to possess the specific structure and exhibit inhibition against ACHN, OS-RC-2, and 786-O human renal cancer cell lines (IC_50_ values from 3.0 to 38 μM) [17]. Further study showed that the new compound isochromophilone D (**1**) could induce cell cycle arrest and even apoptosis in 786-O cells.

#### 3.1.2. Coumarins and Isocoumarins

Coumarins have a benzopyrene structure and their isomers are called isocoumarins [62]. Aspergisocoumrins A-B (**4**–**5**) were two novel isocoumarins derived from the endophytic fungal strain *Aspergillus* sp. HN15-5D, showing significant cytotoxicity against the MDA-MB-435 human breast cancer cell line (IC_50_ values near 5 μM) and possessing inhibitory against the MCF10A human non-cancer breast epithelial cell line (IC_50_ values between 11.3 and 21.4 μM), making an indication that they may have certain safety risks if used in clinical treatment. Aspergisocoumrin A (**4**) showed relatively weak cytotoxicity against human liver cancer cells HepG2 and human lung carcinoma cells H460 (with IC_50_ values > 20 μM) [18]. From another mangrove genus *Fusarium* sp. 2ST2, aspergisocoumrin A (**4**) was also derived and inhibited human lung carcinoma cells, with IC_50_ values of 6.2 ± 0.2 μM [19].

#### 3.1.3. Chromones

With strategic significance in anti-tumor development, chromone and its derivatives have been recognized as the major structural component of various functional organic molecules [63]. The five new chromones were described from four different genera of mangrove associated fungi: *Colletotrichum* sp., *Penicillium* sp., *Cladosporium* sp., and *Fusarium* sp. (5R,7S)-5,7-dihydroxy-2-propyl-5,6,7,8-tetrahydro-4H-chromen-4-one (**6**) from *C. gloeosporioides* exhibited inhibitory activity against A549 cells with an IC_50_ value of 0.094 mM [20]. Penixanthones C-D (**7**–**8**), two chromones from *Penicillium* sp. SCSIO041218 with a unique 6/6/6/5 polycyclic skeleton showed a mild inhibitory effect against K562, MCF-7 (human breast cancer cells), and Huh-7 (human liver cancer cells), with IC_50_ values from 55.2 to 67.5 µM [21]. 7-O-α-D-ribosyl-5-hydroxy-2-propylchromone (**9**) from *Cladosporium* sp. OUCMDZ-302 was cytotoxic to human lung carcinoma cells H1975, with an IC_50_ value of 10.0 µM [22]. 4H-1-benzopyran-4-one-2,3-dihydro-5-hydroxy-8-(hydroxylmethyl)-2-methyl (**10**) was derived from *Fusarium* sp. 2ST2 accompanied by aspergisocoumrin A (**4**), showing the same inhibition against A549 and MDA-MB-435 cells (IC_50_ 5.6 and 3.8 μM, respectively) [19].

#### 3.1.4. Lactones

This group of lactones with six members is characterized by mangrove fungi. Macrolides are abundant in microorganisms and plants, belonging to a class of polyketides with rings of various sizes that are distinguished by the presence of lactone groups [64]. A macrolide obtained both from *Penicillium* sp. (HS-N-27) and (HS-N-29) was identified as brefeldin A (**11**) [23], which is a well-known Golgi-disruptor and the inhibitor of GEFs (the exchange factor for guanine nucleotides) for the ARFs (small GTPases that catalyze ADP-ribosylation) and possessed obvious inhibitory effects on multiple cancer cell lines in the previous study, including lung (A549), colon (HCT-116), glioma (SF-539), melanoma (UACC-62), ovarian (OVCAR-3), renal (SN12C), prostate (PC3), and breast (MCF7) cancer cells, with IC_50_ values lower than 0.1 μM [24]. Similarly, nafuredin B (**12**) obtained from *P. variabile* with co-culture of *Talaromyces aculeatus*, was also found to be a multiple cytotoxic lactones to HeLa (cervical), K562 (renal), HCT-116, HL-60 (myeloid leukemia), A549, and MCF-7 cancer cell lines with IC_50_ values lower than 10 μM [25]. The other four lactones (**13**–**16**) from *A. sydowii* #2B showed cytotoxicities against human prostate cancer VCaP cells, with IC_50_ values from 1.92 to 20.06 μM [26].

#### 3.1.5. Benzoates

Lasiodiplodins are lactones possessing a 14/12-member macrocyclic ring, while 2,4-dihydroxy-6-nonylbenzoate (**17**) and ethyl-2,4-dihydroxy-6-(80-hydroxynonyl)-benzoate (**18**) were detected to be the open-ring structures of lasiodiplodins, characterized as benzoates [27,28]. They were obtained from the same strain *Lasiodiplodia* sp. 318#. The benzoate 2,4-Hihydroxy-6-nonylbenzoate (**17**) was evaluated in vitro against rat pituitary adenoma lines MMQ and GH3 (with IC_50_ values between 5.29 μM and 13.05 μM) [27]. Ethyl-2,4-dihydroxy-6-(80-hydroxynonyl)-benzoate (**18**) showed inhibition against human cancer lines THP1 (monocytic lymphoma), MDA-MB-435, A549, HepG2 (liver), and HCT-116 (with IC_50_ values from 10.13 to 39.74 μM) [28].

#### 3.1.6. Xanthones

Xanthones are dibenzo-pyrone-structured aromatic oxygenated heterocyclic compounds. This molecule may accept different substituents at various places, which are known as privileged scaffolds in the quest for novel medications [65]. There are nine members characterized in this group. Penixanthone A (**19**) from *Penicillium* sp. SYFz-1 showed weak cytotoxicities against the human cancer cell lines, H1975 (lung), MCF-7, and K562, as well as human liver cells HL7702, at a concentration of 30 μM [29]. *Aspergillus* sp. was found to be a promising resource to obtain anti-tumor xanthone metabolites (**20**–**25**, **27**). Versixanthones G-H (**20**–**21**) and L-M (**22**–**23**) from *A. vericolor* were detected to be cytotoxic to HL-60, K562 A549, H1975, MGC803 (stomach), HO-8910 (ovarian), and HCT-116 human cancer cells, but showed low cytotoxicity against human embryonic kidney cells HEK293 [30]. Versixanthones G-H (**20**–**21**) were discovered to be effective DNA Topo I inhibitors to arrest the cell cycle and induce necrosis in MGC803 cells. They exhibited extensive cytotoxicities against five cancer cell lines (HL-60, K562, H1975, MGC803, and HO-8910), with IC_50_ values ranging from 0.4 µM to 16.1 µM. Versixanthones N-O (**24**–**25**) from *A. versicolor* HDN1009 showed multiple cytotoxicities against these cancer cell lines, with IC_50_ values from 1.7 µM to 16.1 µM [31]. Xanthoradone A (**27**) from *A. sydowii* #2B showed strong cytotoxicity against VCaP cells, with IC_50_ values of 4.19 ± 1.02 µM. *Peniophora incarnata* Z4 yielded a tetrahydroxanthone, and incarxanthone B (**26**) was found to inhibit the growth of A375, MCF-7, and HL-60 cell lines, with IC_50_ values of 8.6, 6.5, and 4.9 µM, respectively [32].

#### 3.1.7. Quinones and Benzophenones

Benzophenones may be derived from the corresponding anthraquinones. Penibenzophenones B (**28**), a benzophenone extracted from *Diaporthe* sp. SCSIO 41011 showed cytotoxicity against A549 cell lines (IC_50_ 15.7 µg/mL) [33]. Ten preussomerins, chloropreussomerins A-B (**29–30**), preussomerins A, D, F-H, K, and M (**31**–**36**, **38**), and Ymf 1029E (**37**) from *L. theobromae* ZJ-HQ1 were found to show the cytotoxicity activities towards A549, HepG2, HeLa, and MCF-7 cells (IC_50_ from 2.5 to 83 μM), which is similar to the cytotoxicity to HEK293T cells [34]. Another quinone, the (+)-3,3′-7,7′-8,8′-hexahydroxy-5,5′-dimethyl-bianthra-quinone (**39**) from *A. sydowii* #2B, showed weak cytotoxicity against VCaP cells with an IC_50_ value of 33.36 ± 1.42 μM [26].

#### 3.1.8. Phenols, Phenyl Aldehydes, and Phenolic Acids

Phenyl aldehydes and phenolic acids are phenol derivatives, and there are 15 family members characterized by mangrove fungi. A new benzoic acid, cladoslide A (**40**) from *Cladosporium* sp. HNWSW-1, was discovered to be cytotoxic to the K562 cells with an IC_50_ value of 13.10 ± 0.08 μM [35]. Terphenyllin (**41**) was obtained from *A. candidus* (HS-Y-23), which showed weak cytotoxicity against HeLa cells with an IC_50_ value of 19.0 µM [23]. Asperterphenyllin G (**42**) from *A. candidus* LDJ-5, which was first isolated from a natural source, showed broad activities against the L-02 (liver), MGC-803, HCT-116, BEL-7402 (liver), A549, SH-SY5Y (neuroblastoma), Hela, U87, and HO8910 (ovarian) cancer cell lines, with IC_50_ values lower than 1.7 µM [36]. *Aspergillus* sp. SCSIO41407 yielded a benzophenone flavoglaucin (**43**), showing weak activity against A549 cells with an IC_50_ value of 22.2 μM [37]. Integracins A-B (**44**–**45**) from *Cytospora* sp. displayed significant cytotoxicity against HepG2 lines, with IC_50_ values lower than 10 µM [38]. *Phoma* sp. SYSU-SK-7 yielded colletotric A (**46**) and colletotric B (**47**) and also showed weak cytotoxicity against MAD-MB-435 and A549 cells, with IC_50_ values from 16.82 to 37.73 μM [39]. Seven newly reported prenylated p-terphenyls (**48**–**54**) derived from *A. candidus* LDJ-5 were discovered to possess inhibitory properties against a variety of human cancer cell lines, with IC_50_ values ranging from 0.9 to 29.7 μM [40].

#### 3.1.9. Depsidones

Two depsidones, botryorhodines H and C (**55**–**56**), yielded by *Trichoderma* sp. 307 with the co-culture of *Acinetobacter johnsonii* B2 showed potent cytotoxicity against MMQ and GH3 cell lines, with IC_50_ values from 3.09 to 31.62 μM [41]. 

### 3.2. Terpenoids

Fungi constitute a class of organisms that are particularly appealing for the discovery of terpenoid pathways because they frequently combine their biosynthetic genes [66]. The new terpenoids, including steroids from mangrove fungi in the South China Sea, can be divided into three categories based on their chemical structures and biogenetic pathways: sesquiterpenes, diterpenes, and steroids (Compounds **57**–**70**, Figure 4).

#### 3.2.1. Sesquiterpenes

Sesquiterpenes are the largest group and an excellent source of terpenoids, having a high potential for inhibiting various cancers. A group of well-known natural sesquiterpenoids named eudesmanolides has diverse bioactivities [67]. Two eudesmanolides, 13-hydroxy-3,8,7(11)-eudesmatrien-12,8-olide (**57**) and 13-hydroxy-3,7(11)-eudesmatrien-12,8-olide (**58**), were obtained from *Eutypella* sp. 1–15 exhibiting potent anti-tumor activity against JEKO-1 (human mantle cell lymphoma) and HepG2 cells, with IC_50_ values ranging from 8.4 to 48.4 μM [42]. A novel eudesmane-type sesquiterpenoid penicieudesmol B (**59**) from *Penicillium* sp. J-54 displayed weak cytotoxicity against K-562 cells, with an IC_50_ value of 90.1 µM [43]. *A. ustus* 094102 yielded a drimane sesquiterpenoid ustusolate I (**60**), showing anti-proliferation against CAL-62 (human thyroid cancer) and MG-63 (human osteosarcoma) cells with IC_50_ values of 16.3 and 10.1 µM, respectively [44].

#### 3.2.2. Diterpenes

Diterpenes may be one of the key chemicals in the therapy of cancer, although mangrove fungi have only sometimes produced new bioactive diterpenes [68]. Here, two fungus strains, *Mucor irregularis* and *Eupenicillium* sp. HJ002 were highlighted for the production of nine structurally diverse indole-diterpenes with anti-tumor activity (**61**–**69**), including five novel diterpenes rhizovarins A, B, and E (**61**–**63**) and penicilindoles A-B (**68**–**69**). Rhizovarins A-B (**61**–**62**) possessed a never reported 4,6,6,8,5,6,6,6,6-fused indole-diterpene ring which enduing them with unique chemical properties [45]. The seven bioactive compounds (**61**–**67**) from *M. irregularis* were detected to exhibit significant cytotoxicity against A549 and HL60 cells with IC_50_ values of about 10 µM [45]. Another two diterpenes penicilindoles A-B (**68**–**69**) from *Eupenicillium* sp. HJ002 displayed inhibitory activity against A549 and HepG2 cell lines, with IC_50_ values ranging from 1.5 to 47.2 μM [46].

#### 3.2.3. Steroids

The most significant class of tiny biomolecules, steroids play a variety of cellular functions connected to membrane structure and signaling. *Pseudofusicoccum* sp. J003 yielded a steroid derivative, ergosterol (**70**), displaying significant inhibition effects on HL-60 and SW480 (human colon adenocarcinoma) cells for inhibition rates of 98.68 ± 0.97% and 60.40 ± 4.51% at the concentration of 40 μM [47].

### 3.3. Alkaloids

Alkaloids are a class of compounds containing organic nitrogenous bases that occupy a major part of the secondary metabolites of mangrove fungi. They have been found to have various biological activities [69]. The new alkaloids from mangrove fungi in the South China Sea can be divided into three categories: amines and amides, diketopiperazines, and cytochalasins (Compounds **71**–**109**, Figure 5 and Figure 6).

#### 3.3.1. Amines and Amides

Ascomylactams A (**71**) and C (**72**) were two novel alkaloids from *Didymella* sp. CYSK-4 with a special structure of 12- or 13-membered-ring macrocyclic, displaying moderate anti-proliferative activity against MDA-MB-435, MDA-MB-231, SNB19 (glioma), HCT116, NCI-H460 (lung cancer), and PC-3 cells, with IC_50_ values ranging from 4.2 to 7.8 μM [49]. Further research for ascomylactams A (**71**) showed that it suppressed the growth of A549, NCI-H460, and NCI-H1975 mice tumor xenografts in vivo and arrested the cell cycle through the ROS/Akt/Cyclin D1/Rb pathway both in vivo and in vitro [48]. Fusarisetins E-F (**73**–**74**) were two novel 3-decalinoyltetramic acid derivatives with the structure of a peroxide bridge derived from *Fusarium* sp. 2ST2, showing obvious inhibitory effects on A549 cells, with IC_50_ values of 8.7 to 4.3 μM [19]. The mangrove-derived fungus *Cladosporium* sp. HNWSW-1 yielded two new derivatives containing succinimide, cladosporitin B (**75**) and talaroconvolutin A (**76**), detected to be cytotoxic to BEL-7042, K562, SGC-7901 (human stomach cancer), and Hela cells, with IC_50_ values from 14.9 to 41.7 µM [50].

#### 3.3.2. Diketopiperazines

Diketopiperazine alkaloids, with a disulfide moiety connected to the diketopiperazine ring, play a major role in alkaloids with significant biological properties [70]. Penicisulfuranols A-C (77–79), three new epipolythiodioxopiperazine alkaloids consisting of sulfur atoms on both α- and β-positions of amino acid residues and a 1,2-oxazadecaline core, were obtained from *P. janthinellum* HDN13-309 and were cytotoxic to Hela and HL60 cells, with IC_50_ values ranging from 0.1 to 3.9 μM [51]. Although spirobrocazine C (**80**) and brocazine G (**81**) are two new diketopiperazine alkaloids from the same mangrove-derived fungus *P. brocae* MA-231. Spirobrocazine C (**80**) showed moderate activity against A2780 cells (IC_50_ 59 μM) and brocazine G (**81**) showed strong activity not only to A2780 but also to A2780 CisR cells (IC_50_ 664 and 661 nM) [52]. Saroclazine B (**82**) was a diketopiperazine derivative from *Sarocladium kiliense* HDN11-84 possessing the structure of a free amide found firstly in sulfur-containing aromatic diketopiperazine derivatives, and it was cytotoxic to HeLa cells, with an IC_50_ value of 4.2 μM [53]. A novel trithiodiketopiperazine derivative, adametizine C (**83**), with two dithiodiketopiperazine derivatives (**84**–**85**) from *P. ludwigii* SCSIO 41408 displayed cytotoxicity against human prostate cancer cell line 22Rv1, with IC_50_ values from 13.0 to 13.9 µM, and compound **85** showed significant inhibitory activity against PC-3 cells, with an IC_50_ value of 5.1 µM [54].

#### 3.3.3. Cytochalasins

Cytochalasins are known as a vast class of fungal polyketide-non ribosomal secondary metabolites with a wide variety of biological functions. Possessing a substituted isoindole scaffold joined with a macrocyclic ring produced from a substantially reduced polyketide backbone and an amino acid makes cytochalasins distinctive [71]. Nine cytochalasins from *Chaetomium globosum* kz-19, including phychaetoglobins C-D (**86**–**87**), chaetoglobosins C, E, G, J, V (**88**–**92**), penochalasin J (**93**), and prochaetoglobosin III_ed_ (**94**) displayed moderate cytotoxicities against A549 and HeLa cells, with all the IC_50_ values less than 20 µM [55]. *Phomopsis* sp. QYM-13 yielded four cytochalasins, phomopchalasin E (**95**), cytochalasins U, J (**96**–**97**), and cytochalasin H (**98**), which all showed significant cytotoxicity against MDA-MB-435 cells, with IC_50_ values ranging from 0.2 to 8.2 μM [56]. Moreover, cytochalasins U (**96**) was detected to be cytotoxic to human glioma cell line SNB19, with an IC_50_ value of 6.9 ± 1.4 μM. Two cytochalasins, 12-hydroxylcytochalasin Q (**99**) and zygosporin D (**100**) from *Xylaria arbuscula*, possessed a tetracyclic ring system and displayed significant inhibitory effects on human colorectal adenocarcinoma cell lines HCT15, with IC_50_ values of 13.5 and 13.4 μM, respectively [57]. A new chaetoglobosin with the structure of an unprecedented six-cyclic 6/5/6/5/6/13 fused ring system, penochalasin I (**101**), together with another seven chaetoglobosins penochalasin J (**102**), chaetoglobosins G, F, C, A, and E, and cytoglobosin C (**103**–**108**) were derived from *P. chrysogenum* V11, showing cytotoxicity against MDA-MB-435, SGC-7901 and A549 cells, with IC_50_ values ranging from 3.35 to 38.77 μM [58]. Another new chaetoglobosin, penochalasin K (**109**) with the same special structure of six-cyclic 6/5/6/5/6/13 fused ring and the same fungal strain source, exhibited strong cytotoxicity against MDA-MB-435, SGC-7901, and A549 cells, with IC_50_ values lower than 10 μM [59].

### 3.4. Peptides

In many clinical medication treatments, peptides with molecular weights under 1000 Da can adapt to drug resistance and have fewer hazardous side effects, which may have implications for the ongoing development of novel therapies [72]. In the past seven years, only one cyclic peptide was reported to be an anti-tumor peptide (Compound **110**, Figure 6).

#### Cyclic Peptides

A cyclic peptide, fusaristatin C (**110**), was rapidly separated from *Pseudopithomyces sp.* 1512101 and analyzed using a new strategy of ligand fishing based on PLA2-MNPs (Phospholipase A2-functionalized magnetic nanoparticles) with LC–MS (liquid chromatography–mass spectrometry), exerting a potent inhibitory effect on A549 cells, with IC_50_ values of 10.10 µM [60].

## 4. Discussion

Without a doubt, more and more mangrove secondary metabolites are being discovered and that may be a major source for the creation of novel anti-cancer drugs that can be applied both therapeutically and preventively. Vinca alkaloids are the most notable representatives of plant-derived natural compounds as anticancer weapons, which are frequently utilized as first-line anticancer medications in hematological malignancies [73]. Furthermore, a number of marine isolated targeted compounds, such as Brentuximab2 vedotin (AdcetrisTM), Enfortumab vedotin, and Marizomib, had been used as apoptotic inducers in different cancer types at FDA (the Food and Drug Administration in the US)-approved or treatment phases [74]. More effective natural anti-tumor drugs are expected to emerge.

This review concentrated on one hundred and ten anti-tumor secondary metabolites from thirty-nine mangrove fungus strains belonging to eighteen genera from the South China Sea that have been newly reported over the last seven years. *Penicillium* (23%), *Aspergillus* (23%), and *Lasiodiplodia* (11%) were their main producers and at the structural level, polyketides occupy more than half of the secondary metabolites. Seventy-eight compounds were considered to possess multiple anti-tumor properties, as they exhibited cytotoxicity against more than two tumor cell lines.

It has been proposed that mangrove fungal-derived drugs could be effective weapons for human beings to fight cancer. However, the potential regulatory mechanisms of the vast anti-tumor compounds on tumor microenvironment still need to do in-depth exploration. The bulk of them could not be employed temporarily in tumor diagnosis and treatment due to the lack of reliable clinical research data and studies confirming their biosafety. Additionally, a fungal culture is restricted by special environments or growth factors. Co-cultures of fungi are a popular way to promote the production of metabolites, such as nafuredin B (**12**), botryorhodine H (**55**), and botryorhodine C (**56**), and in the future, multi-omics methods will be richer than the secondary metabolites of mangrove-derived fungi and will be good for screening anti-cancer compounds. Another challenge is figuring out how to produce these secondary metabolites in large quantities.

To date, chemotherapy remains to be regarded as the cornerstone of many adjunctive therapies for cancer. Recently, the design and development of efficient anti-cancer treatment strategies have advanced, and decisions regarding treatment have taken the immunological perspective of chemotherapy into account. Because nucleotides are significant chemicals that must be generated to sustain the state of proliferation in cancer, nucleotide metabolism is regarded as the most crucial link in oncogenesis and progression [75]. On the one hand, targeted regulation of the level of nucleotide metabolism can make tumor cells more sensitive to chemotherapy drugs, mediate anti-tumor response, and enhance the efficacy of chemotherapy and immunotherapy in the treatment of tumors [75,76]. A good anti-tumor impact may also be obtained by altering the amount of nucleotide metabolism, such as by providing exogenous UDP (uridine diphosphate) to modify the tumor microenvironment [77]. This makes immunotherapy tend to have a similar outcome to chemotherapy. On the other hand, chemotherapy stimulates the immune system and triggers tumor cells to undergo immunogenic cell death (ICD) [78]. An important feature of ICD is the extracellular release of ATP from dying cells after apoptosis. For example, daunorubicin, a classic anthracycline anti-tumor drug, induced ATP release into the extracellular space of acute myeloid leukemia cells and was considered a very strong ICD inducer [79]. Another possible drug from the *Agelas mauritianus* sponge, KRN7000, was found to play an anti-tumor role by activating the immune system, having been put into a clinical investigation for many years [80]. In addition, the level of nucleotide release will also be one of the indicators to evaluate the efficacy of anti-tumor drugs in the near future [81].

## 5. Conclusions

Looking forward to the newly developed anti-cancer treatment strategies and anti-cancer metabolite discovery strategies, this review serves as a crucial resource for the research status of mangrove fungal-derived natural products deserving of further development, demonstrating the great medical benefits of mangrove fungal-derived drugs for the treatment of clinical cancers.

## Figures and Tables

**Figure 1 bioengineering-09-00776-f001:**
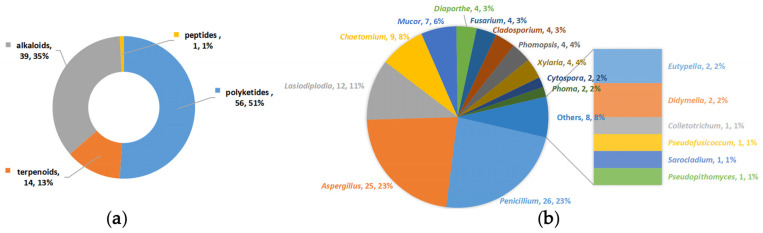
Summarization of anti-tumor secondary metabolites isolated from mangrove-derived fungi. (**a**) shows four types of anti-tumor secondary metabolites. (**b**) shows the genera of fungi producing anti-tumor secondary metabolites.

**Figure 2 bioengineering-09-00776-f002:**
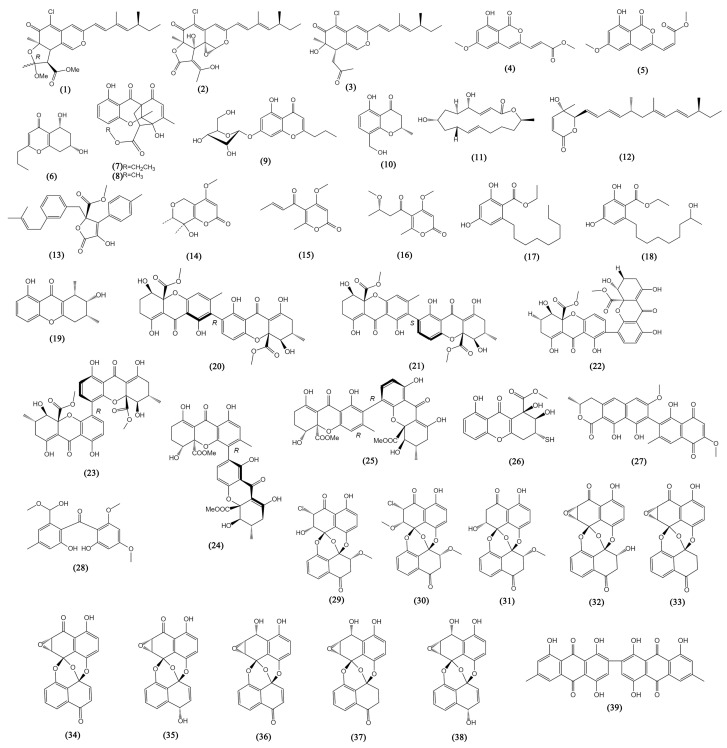
The structures of compounds **1**–**39**.

**Figure 3 bioengineering-09-00776-f003:**
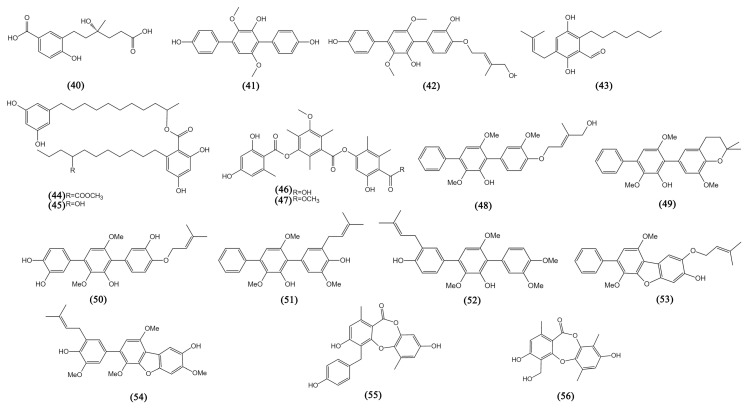
The structures of compounds **40**–**56**.

**Figure 4 bioengineering-09-00776-f004:**
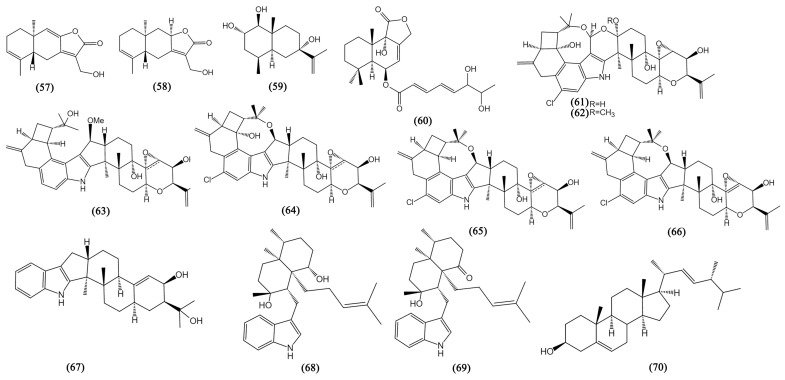
The structures of compounds **57**–**70**.

**Figure 5 bioengineering-09-00776-f005:**
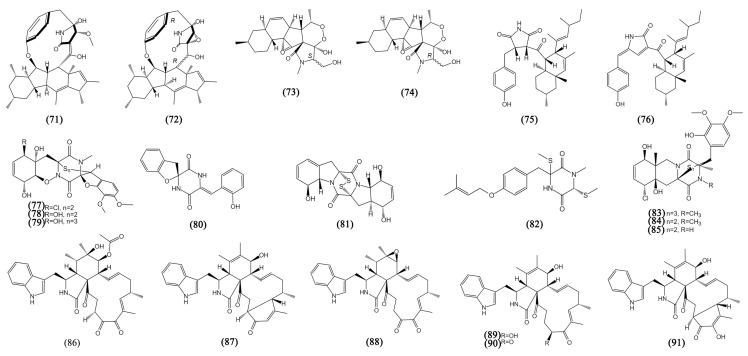
The structures of compounds **71**–**91**.

**Figure 6 bioengineering-09-00776-f006:**
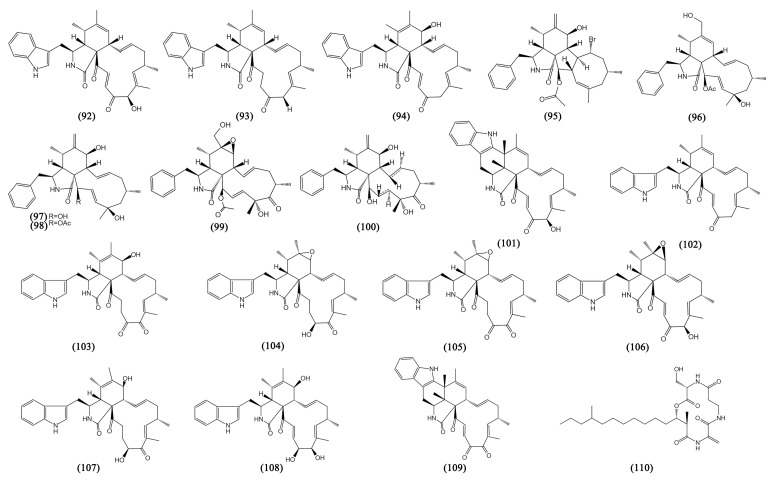
The structures of compounds **92**–**110**.

**Table 1 bioengineering-09-00776-t001:** Anti-tumor activities of secondary metabolites originating from fungal strains in the South China Sea’s mangrove ecosystem.

Type	Source	Compound	Anti-Cancer Type	Cytotoxicity (IC_50_)	Reference
Polyketides					
Azaphilones	*Diaporthe* sp. SCSIO 41011	Isochromophilone D (**1**)	renal cancer	ACHN (14 μM), 786-O (8.9 μM), OS-RC-2 (13 μM)	[17]
		Isochromophilone F (**2**)	renal cancer	ACHN (13 μM), 786-O (10 μM), OS-RC-2 (38 μM)	
		*epi*-isochromophilone II (**3**)	renal cancer	ACHN (4.4 μM), 786-O (3.0 μM), OS-RC-2 (3.9 μM)	
Coumarins and isocoumarins	*Aspergillus* sp. HN15-5D; *Fusarium* sp. 2ST2	Aspergisocoumrin A (**4**)	breast cancer	MDA-MB-435 (5.08 ± 0.88 μM)	[18]
			liver cancer	HepG2 (43.70 ± 1.26 μM)	
			lung cancer	H460 (21.53 ± 1.37 μM); A549 (6.2 ± 0.2 μM)	[18,19]
	*Aspergillus* sp. HN15-5D	Aspergisocoumrin B (**5**)	breast cancer	MDA-MB-435 (5.08 ± 0.88 μM)	[18]
Chromones	*Colletotrichum gloeosporioides*	(5R,7S)-5,7-dihydroxy-2-propyl-5,6,7,8-tetrahydro-4H-chromen-4-one (**6**)	lung cancer	A549 (94.49 μM)	[20]
	*Penicillium* sp. SCSIO041218	Penixanthone C (**7**)	breast cancer	K562 (55.2 μM), MCF-7 (61.1 μM)	[21]
			liver cancer	Huh-7 (67.5 μM)	
		Penixanthone D (**8**)	breast cancer	K562 (56.5 μM), MCF-7 (58.6 μM)	
			liver cancer	Huh-7 (64.2 μM)	
	*Cladosporium* sp. OUCMDZ-302	7-O-α-D-ribosyl-5-hydroxy-2-propylchromone (**9**)	lung cancer	H1975 (10.0 μM)	[22]
	*Fusarium* sp. 2ST2	4H-1-benzopyran-4-one-2,3-dihydro-5-hydroxy-8- (hydroxylmethyl)-2-methyl (**10**)	breast cancer	MDA-MB-435 (3.8 ± 0.3 μM)	[19]
			liver cancer	A549 (5.6 ± 1.3 μM)	
Lactones	*Penicillium* sp. (HS-N-27); *Penicillium* sp. (HS-N-29)	Brefeldin A (**11**)	liver cancer	A549 (0.04 μM)	[23,24]
			colon cancer	HCT-116 (0.03 μM)	
			glioma	SF-539 (0.04 μM)	
			melanoma	UACC-62 (0.02 μM)	
			ovarian cancer	OVCAR-3 (0.03 μM)	
			renal cancer	SN12C (0.09 μM)	
			prostate cancer	PC3 (0.05 μM)	
			breast cancer	MCF7 (0.02 μM)	
	*Penicillium variabile* (co-culture with *Talaromyces aculeatus*)	Nafuredin B (**12**)	cervical cancer	HeLa (5.5 μM)	[25]
			renal cancer	K562 (2.9 μM)	
			colon cancer	HCT-116 (1.4 μM), HL-60 (1.2 μM)	
			lung cancer	A549 (5.1 μM)	
			breast cancer	MCF-7 (9.8 μM)	
	*Aspergillus sydowii* #2B	Butyrolactone-I (**13**)	prostate cancer	VCaP (1.92 ± 0.82 μM)	[26]
		(±)-Pyrenocine S (**14**)	prostate cancer	VCaP (20.06 ± 2.01 μM)	
		Pyrenocine A (**15**)	prostate cancer	VCaP (7.92 ± 0.86 μM)	
		(±)-Pyrenocine E (**16**)	prostate cancer	VCaP (10.13 ± 0.88 μM)	
Benzoates	*Lasiodiplodia* sp. 318^#^	2,4-Dihydroxy-6-nonylbenzoate (**17**)	pituitary adenoma	MMQ (5.29 μM), GH3 (13.05 μM)	[27]
		Ethyl-2,4-dihydroxy-6-(80 -hydroxynonyl)- benzoate (**18**)	breast cancer	MDA-MB-435 (10.13 μM)	[28]
			liver cancer	HepG2 (12.50 μM), A549 (13.31 μM)	
			colon cancer	HCT-116 (11.92 μM)	
			monocytic lymphoma	THP1 (39.74 μM)	
Xanthones	*Penicillium* sp. SYFz-1	Penixanthone A (**19**)	liver cancer	H1975	[29]
			breast cancer	MCF-7	
			renal cancer	K562	
	*Aspergillus vericolor*	Versixanthone G (**20**)	colon cancer	HL-60 (13.4 μM), HCT-116 (16.2 μM)	[30]
			renal cancer	K562 (20.9 μM)	
			lung cancer	A549 (17.8 μM), H1975 (9.8 μM)	
			stomach cancer	MGC803 (4.6 μM),	
			ovarian cancer	HO-8910 (9.6 μM),	
		Versixanthone H (**21**)	colon cancer	HL-60 (6.9 μM), HCT-116 (15.2 μM)	
			renal cancer	K562 (22.1 μM)	
			lung cancer	A549 (19.2 μM), H1975 (5.3 μM)	
			stomach cancer	MGC803 (6.2 μM)	
			ovarian cancer	HO-8910 (6.9 μM)	
		Versixanthone L (**22**)	colon cancer	HL-60 (0.5 μM), HCT-116 (1.2 μM)	
			renal cancer	K562 (1.1 μM)	
			lung cancer	A549 (1.6 μM)	
			stomach cancer	MGC803 (1.1 μM)	
			ovarian cancer	HO-8910 (1.5 μM)	
		Versixanthone M (**23**)	colon cancer	HL-60 (0.9 μM), HCT-116 (0.5 μM)	
			renal cancer	K562 (0.4 μM)	
			lung cancer	A549 (11.7 μM), H1975 (3.5 μM)	
			stomach cancer	MGC803 (0.9 μM)	
			ovarian cancer	HO-8910 (1.4 μM)	
	*Aspergillus versicolor* HDN1009	Versixanthone N (**24**)	colon cancer	HL-60 (2.7 μM)	[31]
			renal cancer	K562 (9.1 μM)	
			lung cancer	H1975 (8.8 μM)	
			stomach cancer	MGC803 (1.7 μM)	
			ovarian cancer	HO-8910 (8.5 μM)	
		Versixanthone O (**25**)	colon cancer	HL-60 (8.7 μM)	
			renal cancer	K562 (16.1 μM)	
			lung cancer	H1975 (8.5 μM)	
			stomach cancer	MGC803 (1.8 μM)	
			ovarian cancer	HO-8910 (6.7 μM)	
	*Peniophora incarnata* Z4	Incarxanthone B (**26**)	melanoma	A375 (8.6 μM)	[32]
			colon cancer	HL-60 (4.9 μM)	
			breast cancer	MCF-7 (6.5 μM)	
	*Aspergillus sydowii* #2B	Xanthoradone A (**27**)	prostate cancer	VCaP (4.19 ± 1.02 μM)	[26]
Benzophenones	*Diaporthe* sp. SCSIO 41011	Penibenzophenone B (**28**)	lung cancer	A549 (15.7 µg/mL)	[33]
Quinones	*Lasiodiplodia theobromae* ZJ-HQ1	Chloropreussomerin A (**29**)	lung cancer	A549 (8.5 ± 0.9 μM)	[34]
			liver cancer	HepG2 (13 ± 1 μM)	
			cervical cancer	HeLa (19 ± 1 μM)	
			breast cancer	MCF-7 (5.9 ± 0.4 μM)	
		Chloropreussomerin B (**30**)	lung cancer	A549 (8.9 ± 0.6 μM)	
			liver cancer	HepG2 (7.7 ± 0.1 μM)	
			cervical cancer	HeLa (27 ± 3 μM)	
			breast cancer	MCF-7 (6.2 ± 0.4 μM)	
		Preussomerin M (**31**)	lung cancer	A549 (36.1 ± 1.2 μM)	
			liver cancer	HepG2 (83 ± 2 μM)	
			breast cancer	MCF-7 (13 ± 1 μM)	
		Preussomerin K (**32**)	lung cancer	A549 (5.4 ± 0.3 μM)	
			liver cancer	HepG2 (3.8 ± 0.9 μM)	
			cervical cancer	HeLa (15 ± 1 μM)	
			breast cancer	MCF-7 (2.5 ± 0.2 μM)	
		Preussomerin H (**33**)	lung cancer	A549 (9.4 ± 0.8 μM)	
			liver cancer	HepG2 (4.4 ± 0.5 μM)	
			cervical cancer	HeLa (19 ± 2 μM)	
			breast cancer	MCF-7 (2.6 ± 0.2 μM)	
		Preussomerin G (**34**)	lung cancer	A549 (6.2 ± 0.1 μM)	
			liver cancer	HepG2 (8.5 ± 0.8 μM)	
			cervical cancer	HeLa (14 ± 1 μM)	
			breast cancer	MCF-7 (4.2 ± 0.6 μM)	
		Preussomerin F (**35**)	lung cancer	A549 (7.7 ± 0.50 μM)	
			liver cancer	HepG2 (3.6 ± 0.6 μM)	
			cervical cancer	HeLa (18 ± 1 μM)	
			breast cancer	MCF-7 (3.1 ± 0.2 μM)	
		Preussomerin D (**36**)	lung cancer	A549 (6.6 ± 0.4 μM)	
			liver cancer	HepG2 (38 ± 2 μM)	
			cervical cancer	HeLa (21 ± 1 μM)	
			breast cancer	MCF-7 (9.8 ± 0.4 μM)	
		Ymf 1029 E (**37**)	lung cancer	A549 (76.2 ± 1.7 μM)	
		Preussomerin A (**38**)	lung cancer	A549 (40.2 ± 1.8 μM)	
			breast cancer	MCF-7 (71 ± 2 μM)	
	*Aspergillus sydowii* #2B	(+)-3,3′-7,7′-8,8′ -hexahydroxy-5,5′-dimethyl-bianthra-quinone (**39**)	prostate cancer	VCaP (33.36 ± 1.42 μM)	[26]
Phenols, phenyl aldehydes, and phenolic acids	*Cladosporium* sp. HNWSW-1	Cladoslide A (**40**)	renal cancer	K562 (13.10 ± 0.08 μM)	[35]
	*Aspergillus candidus (HS-Y-23)*	Terphenyllin (**41**)	cervical cancer	HeLa (19.0 μM)	[23]
	*Aspergillus candidus* LDJ-5	Asperterphenyllin G (**42**)	lung cancer	A549 (0.4 μM)	[36]
			liver cancer	BEL-7402 (6.0 μM)	
			cervical cancer	HeLa (1.7 μM)	
			neuroblastoma	SH-SY5Y (0.6 μM)	
			glioma	U87 (0.9 μM)	
			colon cancer	HCT-116 (0.8 μM)	
			stomach cancer	MGC-803 (1.0 μM)	
			ovarian cancer	HO-8910 (1.3 μM)	
	*Aspergillus* sp. SCSIO41407	Flavoglaucin (**43**)	lung cancer	A549 (22.2 μM)	[37]
	*Cytospora* sp.	Integracin A (**44**)	liver cancer	HepG2 (5.98 ± 0.12 μM)	[38]
		Integracin B (**45**)	liver cancer	HepG2 (9.97 ± 0.06 μM)	
	*Phoma* sp. SYSU-SK-7	Colletotric A (**46**)	lung cancer	A549 (37.73 μM)	[39]
			breast cancer	MAD-MB-435 (37.01 μM)	
		Colletotric B (**47**)	lung cancer	A549 (20.75 μM)	
			breast cancer	MAD-MB-435 (16.82 μM)	
	*Aspergillus candidus* LDJ-5	Prenylterphenyllin F (**48**)	lung cancer	A549 (10.2 μM)	[40]
			liver cancer	BEL-7402 (12.4 μM)	
			cervical cancer	HeLa (8.3 μM)	
			neuroblastoma	SH-SY5Y (10.4 μM)	
			colon cancer	HCT-116 (9.3 μM), HL-60 (7.1 μM)	
			stomach cancer	MGC-803 (11.0 μM)	
		Prenylterphenyllin G (**49**)	lung cancer	A549 (16.3 μM)	
			liver cancer	BEL-7402 (12.6 μM)	
			cervical cancer	HeLa (11.5 μM)	
			neuroblastoma	SH-SY5Y (12.4 μM)	
			colon cancer	HCT-116 (16.9 μM)	
			stomach cancer	MGC-803 (12.5 μM)	
		Prenylterphenyllin H (**50**)	lung cancer	A549 (0.4 μM)	
			cervical cancer	HeLa (2.0 μM)	
			neuroblastoma	SH-SY5Y (0.6 μM)	
			glioma	U87 (13.8 μM)	
			colon cancer	HCT-116 (0.5 μM)	
			stomach cancer	MGC-803 (0.7 μM)	
		Prenylterphenyllin I (**51**)	lung cancer	A549 (14.8 μM)	
			liver cancer	BEL-7402 (11.1 μM)	
			cervical cancer	HeLa (11.4 μM)	
			neuroblastoma	SH-SY5Y (16.7 μM)	
			colon cancer	HCT-116 (14.7 μM)	
			stomach cancer	MGC-803 (14.5 μM)	
		Prenylterphenyllin J (**52**)	lung cancer	A549 (7.6 μM)	
			renal cancer	K562 (15.9 μM)	
			cervical cancer	HeLa (8.5 μM)	
			neuroblastoma	SH-SY5Y (15.6 μM)	
			colon cancer	HCT-116 (6.2 μM)	
			stomach cancer	MGC-803 (8.1 μM)	
		Prenylcandidusin E (**53**)	lung cancer	A549 (19.1 μM)	
			liver cancer	BEL-7402 (14.9 μM)	
			renal cancer	K562 (5.0 μM)	
			cervical cancer	HeLa (14.0 μM)	
			neuroblastoma	SH-SY5Y (17.9 μM)	
			glioma	U87 (10.3 μM)	
			colon cancer	HCT-116 (19.8 μM)	
			stomach cancer	MGC-803 (16.3 μM)	
		Prenylcandidusin G (**54**)	lung cancer	A549 (2.8 μM)	
			liver cancer	BEL-7402 (16.0 μM)	
			cervical cancer	HeLa (10.1 μM)	
			neuroblastoma	SH-SY5Y (2.2 μM)	
			colon cancer	HCT-116 (0.9 μM), HL-60 (3.4 μM)	
			stomach cancer	MGC-803 (1.4 μM)	
Depsidones	*Trichoderma* sp. 307 (co-culture with *Acinetobacter johnsonii* B2)	Botryorhodine H (**55**)	pituitary adenoma	MMQ (3.09 μM), GH3 (3.64 μM)	[41]
		Botryorhodine C (**56**)	pituitary adenoma	MMQ (19.72 μM), GH3 (31.62 μM)	
Terpenoids					
Sesquiterpenoids	*Eutypella* sp. 1–15	13-Hydroxy-3,8,7(11)-eudesmatrien-12,8-olide (**57**)	mantle cell lymphoma	JEKO-1 (8.4 μM)	[42]
			liver cancer	HepG2 (28.5 μM)	
		13-Hydroxy-3,7(11)-eudesmadien-12,8-olide (**58**)	liver cancer	HepG2 (48.4 μM)	
	*Penicillium* sp. J-54	Penicieudesmol B (**59**)	renal cancer	K562 (90.1 μM)	[43]
	*Aspergillus ustus* 094102	Ustusolate I (**60**)	thyroid cancer	CAL-62 (16.3 μM)	[44]
			osteosarcoma	MG-63 (10.1 μM)	
Diterpenes	*Mucor irregularis* QEN-189	Rhizovarin A (**61**)	lung cancer	A549 (11.5 μM)	[45]
			colon cancer	HL60 (9.6 μM)	
		Rhizovarin B (**62**)	lung cancer	A549 (6.3 μM)	
			colon cancer	HL60 (5.0 μM)	
		Rhizovarin E (**63**)	lung cancer	A549 (9.2 μM)	
		Penitrem A (**64**)	lung cancer	A549 (8.4 μM)	
			colon cancer	HL60 (7.0 μM)	
		Penitrem C (**65**)	lung cancer	A549 (8.0 μM)	
			colon cancer	HL60 (4.7 μM)	
		Penitrem F (**66**)	lung cancer	A549 (8.2 μM)	
			colon cancer	HL60 (3.3 μM)	
		3b-hydroxy-4b- desoxypaxilline (**67**)	lung cancer	A549 (4.6 μM)	
			colon cancer	HL60 (2.6 μM)	
	*Eupenicillium* sp. HJ002	Penicilindole A (**68**)	lung cancer	A549 (5.5 μM)	[46]
			liver cancer	HepG2 (1.5 μM)	
			cervical cancer	HeLa (23.3 μM)	
		Penicilindole B (**69**)	lung cancer	A549 (18.6 μM)	
			liver cancer	HepG2 (47.2 μM)	
			cervical cancer	HeLa (20.0 μM)	
Steroids	*Pseudofusicoccum* sp. J003	Ergosterol (**70**)	colon cancer	HL-60, SW480	[47]
Alkaloids					
Amines and amides	*Didymella* sp. CYSK-4	Ascomylactam A (**71**)	breast cancer	MDA-MB-435 (4.9 μM), MDA-MB-231 (5.9 μM)	[48,49]
			glioma	SNB19 (6.8 μM)	
			colon cancer	HCT116 (5.5 μM)	
			lung cancer	NCI-H460 (4.4 μM)	
			prostate cancer	PC-3 (5.7 μM)	
		Ascomylactam C (**72**)	breast cancer	MDA-MB-435 (7.8 μM), MDA-MB-231 (5.1 μM)	
			glioma	SNB19 (7.8 μM)	
			colon cancer	HCT116 (4.2 μM)	
			lung cancer	NCI-H460 (4.4 μM)	
			prostate cancer	PC-3 (7.5 μM)	
	*Fusarium* sp. 2ST2	Fusarisetin E (**73**)	lung cancer	A549 (8.7 μM)	[19]
		Fusarisetin F (**74**)	lung cancer	A549 (4.3 μM)	
	*Cladosporium* sp. HNWSW-1	Cladosporitin B (**75**)	liver cancer	BEL-7042 (29.4 ± 0.35 μM)	[50]
			renal cancer	K562 (25.6 ± 0.47 μM)	
			stomach cancer	SGC-7901 (41.7 ± 0.71 μM)	
		Talaroconvolutin A (**76**)	cervical cancer	HeLa (14.9 ± 0.21 μM)	
			liver cancer	BEL-7042 (26.7 ± 1.1 μM)	
Diketpiperazines	*Penicillium janthinellum* HDN13-309	Penicisulfuranol A (**77**)	cervical cancer	HeLa (0.5 μM)	[51]
			colon cancer	HL-60 (0.1 μM)	
		Penicisulfuranol B (**78**)	cervical cancer	HeLa (3.9 μM)	
			colon cancer	HL-60 (1.6 μM)	
		Penicisulfuranol C (**79**)	cervical cancer	HeLa (0.3 μM)	
			colon cancer	HL-60 (1.2 μM)	
	*Penicillium brocae* MA-231	Spirobrocazine C (**80**)	ovarian cancer	A2780 (59 μM)	[52]
		Brocazine G (**81**)	ovarian cancer	A2780 (0.664 μM), A2780 CisR (0.661 μM)	
	*Sarocladium kiliense* HDN11-84	Saroclazine B (**82**)	cervical cancer	HeLa (4.2 μM)	[53]
	*Penicillium ludwigii* SCSIO 41408	Adametizine C (**83**)	prostate cancer	22Rv1 (13.9 μM)	[54]
		Compound (**84**)	prostate cancer	22Rv1 (13.0 μM)	
		Compound (**85**)	prostate cancer	22Rv1 (13.6 μM), PC-3 (5.1 μM)	
Cytochalasans	*Chaetomium globosum* kz-19	Phychaetoglobin C (**86**)	cervical cancer	HeLa (16.1 ± 0.3 μM)	[55]
			lung cancer	A549 (22.3 ± 0.4 μM)	
		Phychaetoglobin D (**87**)	cervical cancer	HeLa (9.2 ± 0.3 μM)	
			lung cancer	A549 (13.7 ± 0.2 μM)	
		Chaetoglobosin C (**88**)	cervical cancer	HeLa (10.5 ± 0.1 μM)	
			lung cancer	A549 (7.6 ± 0.2 μM)	
		Chaetoglobosin E (**89**)	cervical cancer	HeLa (7.5 ± 0.2 μM)	
			lung cancer	A549 (12.3 ± 0.3 μM)	
		Chaetoglobosin G (**90**)	cervical cancer	HeLa (3.7 ± 0.3 μM)	
			lung cancer	A549 (7.3 ± 0.5 μM)	
		Chaetoglobosin V (**91**)	cervical cancer	HeLa (3.8 ± 0.3 μM)	
			lung cancer	A549 (11.0 ± 0.2 μM)	
		Chaetoglobosin J (**92**)	lung cancer	A549 (13.4 ± 0.1 μM)	
		Penochalasin J (**93**)	lung cancer	A549 (14.9 ± 0.2 μM)	
		Prochaetoglobosin III_ed_ (**94**)	cervical cancer	A549 (17.3 ± 0.3 μM)	
			lung cancer	HeLa (12.2 ± 0.1 μM)	
	*Phomopsis* sp. QYM-13	Phomopchalasin E (**95**)	breast cancer	MDA-MB-435 (7.4 ± 1.5 μM)	[56]
		Cytochalasin U (**96**)	breast cancer	MDA-MB-435 (8.2 ± 0.9 μM)	
			glioma	SNB19 (6.9 ± 1.4 μM)	
		Cytochalasin J (**97**)	breast cancer	MDA-MB-435 (4.9 ± 0.6 μM)	
		Cytochalasin H (**98**)	breast cancer	MDA-MB-435 (0.2 ± 0.1 μM)	
	*Xylaria arbuscula*	12-hydroxylcytochalasin Q (**99**)	colorectal adenocarcinoma	HCT15 (13.5 μM)	[57]
		Zygosporin D (**100**)	colorectal adenocarcinoma	HCT15 (13.4 μM)	
	*Penicillium chrysogenum* V11	Penochalasin I (**101**)	breast cancer	MDA-MB-435 (7.55 ± 0.71 μM)	[58]
			stomach cancer	SGC-7901 (7.32 ± 0.68 μM)	
			lung cancer	A549 (16.13 ± 0.82 μM)	
		Penochalasin J (**102**)	breast cancer	MDA-MB-435 (36.68 ± 0.90 μM)	
			stomach cancer	SGC-7901 (37.70 ± 1.30 μM)	
			lung cancer	A549 (35.93 ± 0.66 μM)	
		Chaetoglobosin G (**103**)	breast cancer	MDA-MB-435 (38.77 ± 0.65 μM)	
			stomach cancer	SGC-7901 (25.86 ± 0.84 μM)	
			lung cancer	A549 (27.63 ± 0.45 μM)	
		Chaetoglobosin F (**104**)	breast cancer	MDA-MB-435 (37.77 ± 0.41 μM)	
			stomach cancer	SGC-7901 (26.53 ± 0.56 μM)	
			lung cancer	A549 (27.72 ± 0.81 μM)	
		Chaetoglobosin C (**105**)	breast cancer	MDA-MB-435 (19.97 ± 1.03 μM)	
			stomach cancer	SGC-7901 (15.36 ± 0.89 μM)	
			lung cancer	A549 (17.82 ± 0.85 μM)	
		Chaetoglobosin A (**106**)	breast cancer	MDA-MB-435 (37.56 ± 0.95 μM)	
			stomach cancer	SGC-7901 (7.48 ± 1.01 μM)	
			lung cancer	A549 (6.56 ± 0.67μM)	
		Chaetoglobosin E (**107**)	lung cancer	A549 (36.63 ± 0.45 μM)	
		Cytoglobosin C (**108**)	breast cancer	MDA-MB-435 (12.58 ± 0.90 μM)	
			stomach cancer	SGC-7901 (8.15 ± 0.64 μM)	
			lung cancer	A549 (3.35 ± 0.47 μM)	
	*Penicillium chrysogenum* V11	Penochalasin K (**109**)	breast cancer	MDA-MB-435 (4.65 ± 0.45 μM)	[59]
			stomach cancer	SGC-7901 (5.32 ± 0.58 μM)	
			lung cancer	A549 (8.73 ± 0.62 μM)	
Peptides					
Cyclic peptides	*Pseudopithomyces* sp. 1512101	Fusaristatin C (**110**)	lung cancer	A549 (10.10 μM)	[60]

## Data Availability

No new data were created or analyzed in this study. Data sharing is not applicable to this article.

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
