# Peer review of "Anti-Tumor Secondary Metabolites Originating from Fungi in the South China Sea’s Mangrove Ecosystem"

_bioengineering, 2022, doi:10.3390/bioengineering9120776_

Round 1
Reviewer 1 Report
This paper is clear and interesting. My only suggestion is to insert a new short paragraph on the role of nucleotides in the immune response and in their pro-tumor / anti-tumor action.
As reference, the authors could use PMID: 34604232
Author Response
Response for reviewers for manuscript bioengineering-1997373
First of all, we would like to thank you for the time and effort you have spent reviewing our article (Manuscript Number: bioengineering-1997373). We are pleased to note that you have found our work interesting and also pointed out some problems to help us improve the quality of our article. All authors seriously discussed all comments. Based on your comments, we have done our best to modify our manuscript to meet the requirements of your journal. In this revision, changes to the manuscript in the document are highlighted in red text. Revised portion are marked in red in the paper. The main corrections in the paper and the response to the reviewer' s comments are as flowing:
-My only suggestion is to insert a new short paragraph on the role of nucleotides in the immune response and in their pro-tumor / anti-tumor action.
Response: This is really a very helpful advice. We agreed that the role of RNA, especially the non-coding RNAs, would become a research focus on the cancer treatment in the near future. The description of RNA in the immune response were added to the discussion as below (Paragraph 2, Pages 16-17, Lines 376-386):
Modulating the elements of tumor immune microenvironment is a promising therapeutic approach against cancer. RNA molecules expressed in tumor cells, including microRNAs and long non-coding RNAs, influence suppressive immunological microenvironments either directly or indirectly. It is clinically proven that RNA-mediated therapies when used in combination with other therapies may have a good application prospect in cancer treatment [75]. For instance, microRNAs, whose biogenesis could be manipulated by some anti-tumor drugs, have been shown to be closely related to drug therapy for tumors according to the clinical researches [76]. The best example of mangrove-related compounds is SZ-685C, the marine anthraquinone derivative from Halorosellinia sp. (No. 1403), which is discovered to decrease miR-200c and induce the MMQ cell apoptosis [77].
References:
- Pandey, P.R.; Young, K.H.; Kumar, D.; Jain, N. RNA-mediated immunotherapy regulating tumor immune microenvironment: next wave of cancer therapeutics. Mol. Cancer 2022, 21. 10.1186/s12943-022-01528-6.
- Arghiani, N.; Shah, K. Modulating microRNAs in cancer: Next-generation therapies. Cancer Biol. Med. 2021, 1. 10.20892/j.issn.2095-3941.2021.0294.
- Chen, C.H.; Xiao, W.W.; Jiang, X.B.; Wang, J.W.; Mao, Z.G.; Lei, N.; Fan, X.; Song, B.B.; Liao, C.X.; Wang, H.J.; She, Z.G.; Zhu, Y.H. A novel marine drug, SZ-685C, induces apoptosis of MMQ pituitary tumor cells by downregulating miR-200c. Curr. Med. Chem. 2013, 20, 2145-2154. 10.2174/0929867311320160007.
Reviewer 2 Report
Thank you for the opportunity to review this manuscript.
In this narrative review, Luo and collaborators have summarized the available literature regarding the availability of anti-tumor secondary metabolites from mangrove-derived fungi in the South China Sea.
The manuscript is well-written and seems scientifically sound.
Several suggestions for revision are listed below:
*In my view, the abstract is too short and might not attract readers to check the full text of your paper.
*Explain all abbreviations at their first use, e.g., what is OSMAC?
*the quality of the figures needs to be improved - you must import them differently from Excel or other programs to keep their quality
*table 1 - abbreviations not explained. It is unclear by the name of cell lines against which cancer types these compounds are effective. Add another column and add the cancer types.
*other currently antineoplastic compounds have been derived from alkaloids and other of the mentioned molecules - you should briefly mention them. See the following book chapter regarding this topic:
https://www.sciencedirect.com/science/article/pii/B9780128178904000068
*The discussions should be improved and presented separately from the conclusions
*try and reduce the similarity index, some parts of the paper need to be rephrased

Author Response
Response for reviewers for manuscript bioengineering-1997373
First of all, we would like to thank you for the time and effort you have spent reviewing our article (Manuscript Number: bioengineering-1997373). We are pleased to note that you have found our work interesting and also pointed out some problems to help us improve the quality of our article. All authors seriously discussed all comments. Based on your comments, we have done our best to modify our manuscript to meet the requirements of your journal. In this revision, changes to the manuscript in the document are highlighted in red text. Revised portion are marked in red in the paper. The main corrections in the paper and the response to the reviewer' s comments are as flowing:
- The abstract is too short and might not attract readers to check the full text of your paper.
Response: Thank you for your advice. The abstract was altered in the revised manuscript.
- Explain all abbreviations at their first use, e.g., what is OSMAC?
Response: Thanks for the reminder. OSMAC means "one strain, many compounds". The explanations of other abbreviations were added at their first use in the revised manuscript.
- the quality of the figures needs to be improved - you must import them differently from Excel or other programs to keep their quality
Response: Sorry for the mistake. The figures were improved in the revised manuscript, including highlighting fonts on figures, replacing pictures that were unclear, and altering positions of compounds for better visibility.
- table 1 - abbreviations not explained. It is unclear by the name of cell lines against which cancer types these compounds are effective. Add another column and add the cancer types.
Response: It is a really wonderful suggestion. The abbreviation of IC50 were explained in the text. The column of anti-cancer types was added in the revised table and the cell lines corresponding to cancer types were clearly shown.
- other currently anti-neoplastic compounds have been derived from alkaloids and other of the mentioned molecules - you should briefly mention
Response: We appreciate the reviewers' advice. Other natural anti-cancer compounds, including first-line clinical drugs and FDA approved drugs were added in the discussion as below (Paragraph 1, Page 16, Lines 368-375):
Without a doubt, more and more mangrove secondary metabolites are being discovered and that may be a major source for the creation of novel anti-cancer drugs that can be applied both therapeutically and preventively. Vinca alkaloids are the most notable representatives of plant-derived natural compounds as anticancer weapons, which are frequently utilized as first-line anticancer medications in hematological malignancies [73]. A number of marine isolated targeted compounds, like Brentuximab2 vedotin (AdcetrisTM), Enfortumab vedotin and Marizomib, had been used as apoptotic inducers in different cancer types at FDA (Food and Drug Administration in US) approved or treatment phase [74]. More effective natural anti-tumor drugs are being expected to emerge.
References:
- Mariagăman, A.; Egbuna, C.; Găman, M. Chapter 6 - Natural bioactive lead compounds effective against haematological malignancies. In Phytochemicals as Lead Compounds for New Drug Discovery, 2020, pp. 95-115. 10.1016/B978-0-12-817890-4.00006-8.
- Chaudhry, G.E.; Md, A.A.; Sung, Y.Y.; Sifzizul, T. Cancer and apoptosis: The apoptotic activity of plant and marine natural products and their potential as targeted cancer therapeutics. Front. Pharmacol. 2022, 13, 842376. 10.3389/fphar.2022.842376.
- The discussions should be improved and presented separately from the conclusions
Response: Thanks for the suggestion. The discussions were altered and the conclusions were presented separately.
- try and reduce the similarity index, some parts of the paper need to be rephrased
Response: Thanks for your advice, it is an important point. The similarity index were reduced and most of the fungal strains' names were replaced as abbreviations. Some paragraphs in the manuscript were also rephrased.
Reviewer 3 Report
This is interesting and well written review focused on mangrove fungi-derived secondary metabolites with anticancer potential. It is based on 44 publication and described 110 compounds. The collected data are summarized in form of Table and the structures of the components are included to the study.
I have some editorial suggestions before acceptance for publication:
1. What is the main question addressed by the research? The manuscript summarizes the current state of knowledge on mangrove fungi-derived secondary metabolites with anticancer potential
2. Do you consider the topic original or relevant in the field? Does it address a specific gap in the field? The topic is relevant and original
3. What does it add to the subject area compared with other published material? It includes the data from recent articles (from 2016-2022). The information on has been updated
4. What specific improvements should the authors consider regarding the methodology? What further controls should be considered? Methodology is correct
5. Are the conclusions consistent with the evidence and arguments presented and do they address the main question posed? Conclusion is correct
6. Are the references appropriate? References are appropriate
7. Please include any additional comments on the tables and figures. Fonts on figures (e.g. for the number of compounds) should be larger to better visibility. All Figures should be cited in the text.
Author Response
Response for reviewers for manuscript bioengineering-1997373
First of all, we would like to thank you for the time and effort you have spent reviewing our article (Manuscript Number: bioengineering-1997373). We are pleased to note that you have found our work interesting and also pointed out some problems to help us improve the quality of our article. All authors seriously discussed all comments. Based on your comments, we have done our best to modify our manuscript to meet the requirements of your journal. In this revision, changes to the manuscript in the document are highlighted in red text. Revised portion are marked in red in the paper. The main corrections in the paper and the response to the reviewer' s comments are as flowing:
- What is the main question addressed by the research?
Response: We think this is a very important question. This review includes 110 anti-tumor secondary metabolites from 39 mangrove fungal strains belonging to 18 genera in the South China Sea reported during 2016 and 2022, serving as a crucial resource for the research status of potential anticancer natural products.
- Do you consider the topic original or relevant in the field? Does it address a specific gap in the field? The topic is relevant and original
Response: Thanks for the question. As the description in our introduction and discussion, drug therapy is still an indispensable part of anti-tumor therapy. However, the safety and resistance of cytotoxic drugs are unavoidable clinical concerns. More effective natural anti-tumor drugs are being expected to emerge and this review may provide a new direction in the search for drugs.
- What does it add to the subject area compared with other published material? It includes the data from recent articles (from 2016-2022). The information on has been updated
- What specific improvements should the authors consider regarding the methodology? What further controls should be considered? Methodology is correct
- Are the conclusions consistent with the evidence and arguments presented and do they address the main question posed? Conclusion is correct
- Are the references appropriate? References are appropriate
- Please include any additional comments on the tables and figures. Fonts on figures (e.g. for the number of compounds) should be larger to better visibility. All Figures should be cited in the text.
Response: We appreciate the reviewers' advice. The figures were improved in the revised manuscript, including highlighted fonts on figures and altered composition of compounds for better visibility. Figures were also cited in the revised text.
Round 2
Reviewer 1 Report
Because the review of the authors is focused on metabolites, I suggested to insert a brief chapter on the role of nucleotides (monophosphate, diphosphate or triphosphate) in the immune response and their pro-tumoral/anti-tumoral actions and not on RNA (non coding RNA, microRNA ...). Further, I suggested the authors a specific reference (PMID: 34604232, Vecchio et al., Metabolites profiling of melanoma interstitial fluid reveals uridine diphosphate as potent immune modulator capable to limit the tumor growth, 2021) in which the role of nucleotides in the tumor context has been addressed. Unfortunately, the authors didn't take any of my suggestions in consideration.
Author Response
-Because the review of the authors is focused on metabolites, I suggested to insert a brief chapter on the role of nucleotides (monophosphate, diphosphate or triphosphate) in the immune response and their pro-tumoral/anti-tumoral actions and not on RNA (non coding RNA, microRNA ...). Further, I suggested the authors a specific reference (PMID: 34604232, Vecchio et al., Metabolites profiling of melanoma interstitial fluid reveals uridine diphosphate as potent immune modulator capable to limit the tumor growth, 2021) in which the role of nucleotides in the tumor context has been addressed. Unfortunately, the authors didn't take any of my suggestions in consideration.
Response: We are deeply sorry for our earlier misunderstanding. Thanks to the reviewers' comments and recommendations, we have adjusted the description of nucleotides in the immune response which were added to the discussion as below (Paragraph 4, Pages 17, Lines 409-430):
To date, chemotherapy remains to be regarded as the cornerstone of many adjunctive therapies for cancer. Recently, the design and development of efficient anti-cancer treatment strategies have recently advanced, and decisions regarding treatment have taken the immunological perspective of chemotherapy into account. Because nucleotides are significant chemicals that must be generated to sustain the state of proliferation in cancer, nucleotide metabolism is regarded as the most crucial link in oncogenesis and progression [75]. On the one hand, targeted regulation of the level of nucleotide metabolism can make tumor cells more sensitive to chemotherapy drugs, mediate anti-tumor response, and enhance the efficacy of chemotherapy and immunotherapy in the treatment of tumors [75,76]. A good anti-tumor impact may also be obtained by altering the amount of nucleotide metabolism, such as by providing exogenous UDP (uridine diphosphate) to modify the tumor microenvironment [77]. This makes immunotherapy tend to have a similar outcome to chemotherapy. On the other hand, chemotherapy stimulates the immune system and triggers tumor cells to undergo immunogenic cell death (ICD) [78]. An important feature of ICD is the extracellular release of ATP from dying cells after apoptosis. For example, daunorubicin, a classic anthracycline anti-tumor drug, induced ATP release into the extracellular space of acute myeloid leukemia cells and was considered a very strong ICD inducer [79]. Another possible drug from the Agelas mauritianus sponge, KRN7000, was found to play an anti-tumor role by specially activating the immune system, having been put into clinical investigation for many years [80]. In addition, the level of nucleotide release will also be one of the indicators to evaluate the efficacy of anti-tumor drugs in the near future [81].
References:
- Wu, H.; Gong, Y.; Ji, P.; Xie, Y.; Jiang, Y.; Liu, G. Targeting nucleotide metabolism: a promising approach to enhance cancer immunotherapy. J. Hematol. Oncol. 2022, 15. 10.1186/s13045-022-01263-x.
- Zanoni, M.; Pegoraro, A.; Adinolfi, E.; De Marchi, E. Emerging roles of purinergic signaling in anti-cancer therapy resistance. Front. Cell. Dev. Biol. 2022, 10. 10.3389/fcell.2022.1006384.
- Vecchio, E.; Caiazza, C.; Mimmi, S.; Avagliano, A.; Iaccino, E.; Brusco, T.; Nisticò, N.; Maisano, D.; Aloisio, A.; Quinto, I.; Renna, M.; Divisato, G.; Romano, S.; Tufano, M.; D Agostino, M.; Vigliar, E.; Iaccarino, A.; Mignogna, C.; Andreozzi, F.; Mannino, G.C.; Spiga, R.; Stornaiuolo, M.; Arcucci, A.; Mallardo, M.; Fiume, G. Metabolites Profiling of Melanoma Interstitial Fluids Reveals Uridine Diphosphate as Potent Immune Modulator Capable of Limiting Tumor Growth. Front. Cell. Dev. Biol. 2021, 9. 10.3389/fcell.2021.730726.
- Solari, J.I.G.; Filippi-Chiela, E.; Pilar, E.S.; Nunes, V.; Gonzalez, E.A.; Figueiró, F.; Andrade, C.F.; Klamt, F. Damage-associated molecular patterns (DAMPs) related to immunogenic cell death are differentially triggered by clinically relevant chemotherapeutics in lung adenocarcinoma cells. BMC Cancer 2020, 20. 10.1186/s12885-020-06964-5.
- Ocadlikova, D.; Iannarone, C.; Redavid, A.R.; Cavo, M.; Curti, A. A Screening of Antineoplastic Drugs for Acute Myeloid Leukemia Reveals Contrasting Immunogenic Effects of Etoposide and Fludarabine. International Journal of Molecular Sciences 2020, 21, 6802. 10.3390/ijms21186802.
- Schneiders, F.L.; Scheper, R.J.; von Blomberg, B.M.E.; Woltman, A.M.; Janssen, H.L.A.; van den Eertwegh, A.J.M.; Verheul, H.M.W.; de Gruijl, T.D.; van der Vliet, H.J. Clinical experience with α-galactosylceramide (KRN7000) in patients with advanced cancer and chronic hepatitis B/C infection. Clin. Immunol. 2011, 140, 130-141. 10.1016/j.clim.2010.11.010.
- Wu, C.; Wang, H.; Lin, M.; Chu, L.; Liu, R. Radiolabeled nucleosides for predicting and monitoring the cancer therapeutic efficacy of chemodrugs. Curr. Med. Chem. 2012, 19, 3315. 10.1186/s13045-022-01263-x.
Again, we sincerely thank you for your help.